# The Dramatic Consequences of an Accidental Ligation of the Celiac Trunk during Surgery Performed on a Child with Neuroblastoma

**DOI:** 10.3390/ijerph18041841

**Published:** 2021-02-14

**Authors:** Patrycja Sosnowska-Sienkiewicz, Danuta Januszkiewicz-Lewandowska, Przemysław Mańkowski

**Affiliations:** 1Department of Pediatric Surgery, Traumatology and Urology, Poznan University of Medical Sciences, Szpitalna Street 27/33, 60-572 Poznan, Poland; mankowskip@ump.edu.pl; 2Department of Pediatric Oncology, Hematology and Transplantology, Poznan University of Medical Sciences, Szpitalna Street 27/33, 60-572 Poznan, Poland; danuta.januszkiewicz@ump.edu.pl; 3Department of Medical Diagnostics, Dobra Street 38a, 60-595 Poznan, Poland

**Keywords:** celiac trunk, child, neuroblastoma, surgery, oncology

## Abstract

Neuroblastoma is the most common extra-cranial solid tumor in infants and young children, and accounts for approximately 8–10% of all childhood cancers. The International Neuroblastoma Staging System (The International Neuroblastoma Risk Group Staging System (INRGSS)) is based on the age of patient and preoperative imaging, with attention paid to whether the primary tumor is affected by one or more of specific Image-Defined Risk Factors (IDRFs). Patients are classified into the following groups: locoregional L1 and L2 (absent or present IDRFs respectively), M stage (a disseminated form of neuroblastoma) and Ms (the stage present in children younger than 18 months of age with the disease spread to the bone marrow and/or liver, and/or skin). This publication is aimed to present an unexpected complication associated with an accidental ligation of the celiac trunk during resection of a neuroblastoma tumor in a 2.5-year-old boy after initial chemotherapy, initially with vascular IDRFs, stage L2. The consequences of this complication were pancreatic and spleen ischemia and necrosis, and ischemia and perforation of the common bile duct, gallbladder, stomach, and duodenum. Despite detailed diagnostic imaging (computed tomography, magnetic resonance), the presence of vascular IDRFs may result in an unexpected complication in the surgical treatment of neuroblastoma in children.

## 1. Introduction

Neuroblastoma is the most common extra-cranial solid tumor in infants and young children, and accounts for approximately 8–10% of all childhood cancers [1]. Of note, 90% of cases are diagnosed before the age of 5 [2].

The clinical presentation of the tumor is affected by many factors. It depends on the tumor’s localization, size, stage, symptoms of catecholamine secretion, and the presence of paraneoplastic syndromes [1,2,3]. About 65% of tumors are located in the abdomen, with half of those forming in the medulla of the adrenal gland. They can also occur in the neck, chest, pelvis, or other locations [1,4]. A large number of patients are asymptomatic at the moment of diagnosis. Some have abdominal distension, respiratory distress, constipation, occlusion and/or dysphagia. Secretion of catecholamines may result in hypertension and tachycardia. Paraneoplastic syndromes may cause diarrhea, encephalomyelitis, sensory neuropathy, or opsoclonus-myoclonus syndrome [5].

To minimize complications associated with the surgical procedure, the Image-Defined Risk Factors (IDRFs) were introduced in 2009. They have been designed to guide surgical management of neuroblastoma at diagnosis, in particular, to indicate whether a biopsy or attempted resection is recommended as the first surgical procedure. The same system of risk factors can be applied later during treatment and follow-up [6,7]. Summarizing, it can be said that IDRFs are surgical risk factors and are identified by imaging, based on radiological criteria. For example, abdominal and pelvic IDRFs include: tumor-infiltrating porta hepatis or hepatoduodenal ligament, encasing branches of superior mesenteric artery at mesenteric root, encasing origin of celiac axis or superior mesenteric artery, invading one or both renal pedicles, encasing aorta or vena cava, encasing iliac vessels, pelvic tumor crossing sciatic notch [7].

In the International Neuroblastoma Staging System (The International Neuroblastoma Risk Group Staging System (INRGSS)), locoregional tumors are staged L1 or L2 based on the absence or presence of one or more of the Image-Defined Risk Factors (IDRFs), respectively assessed at the moment of diagnosis. Metastatic tumors are defined as stage M, except for stage MS, in which metastases are confirmed in children younger than 18 months of age to the skin, liver, and/or bone marrow [6,7].

Other factors which influence the prognosis and treatment results are the age of patient, the histopathological type of the tumor, and the presence of genetic markers (MYCN amplification, tumor cell ploidy, the presence of 11q aberration) [6,7].

This publication is aimed to present an unexpected complication associated with an accidental ligation of the celiac trunk (celiac artery, celiac axis) during resection of a neuroblastoma tumor in a 2.5-year-old boy after initial chemotherapy, initially with vascular IDRFs and stage L2.

## 2. Case Report

A 2.5-year-old boy was admitted to the hospital because he had been lethargic for about two weeks, had experienced chest and abdominal pain, and had been vomiting.

The boy had been born from a second pregnancy at 39 weeks of gestation, with a birth weight of 3790 g and an APGAR (Appearance, Pulse, Grimace, Activity, Respiration) score of 10. The pregnancy had gone without any complications. Our patient had a negative family history concerning cancer diseases. His parents, older brother, and the closest family were healthy.

On admission to the oncology department, the patient was in a good general condition and showed no abnormality in a physical examination. Imaging examinations—ultrasonography and computed tomography (USG, CT) anteriorly from the right kidney and the thoracolumbar spine showed a solid mass of the tumor, ranging from Th11 to L3 and undergoing significant contrast enhancement. Within the tumor, calcifications were visualized. The dimensions of the lesion were determined to be 4.9 × 4.6 × 7.3 cm^3^. The lesion adhered to the pancreas, the right adrenal gland, the liver’s visceral surface, the right kidney, the duodenum, the right leg of the diaphragm, and the right psoas muscle, showing the features of a neoplastic infiltrate. The tumor modelled the inferior vena cava (VCI), which was about 2.5 cm away from the spine and segmentally narrowed to 4 mm. The tumor was in close proximity to large vessels—aorta, celiac trunk, common hepatic artery, superior mesenteric artery, VCI, right and left renal veins, splenic and portal vein. (Figure 1). A neuroblastoma lesion was suspected. Three IDRFs have been identified—the tumor-encased aorta and vena cava, the origin of the celiac axis, and branches of the superior mesenteric artery at mesenteric roots.

Computed tomography (CT), magnetic resonance (MR), and MIBG (metaiodobenzylguanidine) scintigraphy, as well as bone marrow aspirate and biopsy, excluded the presence of metastases in the lungs, liver, skeletal system, bone marrow, and central nervous system. According to the The International Neuroblastoma Risk Group Staging System (INRGSS), the tumor was classified as L2 (IDRFs present). Subsequently, an open biopsy of the tumor was performed. (The histopathological examination showed a neuroblastoma of a differentiating type, partially maturing, with medium density tumor tissue, with a low mitotic index and favorable histology [FH]. In the immunohistochemical examination, Neuron-specific enolase (NSE) [+], Synaptophysin [−], CD56 [+] and S-100p [+] were analyzed. Ki-67 was positive in 20% of the cells. N-myc(-) gene amplification was absent. The vascular port was implemented and chemotherapy was applied. After two cycles of Vepeside and Carboplatin, a check-up CT was done in which the decrease of the tumor mass was detected (3.8 × 3.9 × 6.8 cm^3^). The tumor volume reduction exceeded 35% (Figure 1).

It was decided to remove the tumor and to continue with further postoperative chemo- and/or radiotherapy. 

Laparotomy and non-radical resection of the tumor in the retroperitoneal space were performed. The tumor was dissected from the inferior vena cava, renal veins, aorta, splenic vessels, and adjacent organs like the pancreas, liver, and duodenum. In order to allow for the resection, it was necessary to ligate the vessel with a diameter of 4 mm, entering the lesion centrally (Figure 1). It was not possible to dissect the tumor from the other structures; therefore, that part was not removed. For the first 24 h after the surgery, the patient stayed in the intensive care unit, and then he was transferred to the surgical ward. The patient was receiving standard low-molecular-weight heparin therapy in a prophylactic dose. At 48 h after the surgery, an abdominal cavity ultrasound was performed due to abdominal pain. The USG did not visualize the flow through the splenic artery; the splenic vein was collapsed. There was flow through the common hepatic artery, portal vein, and other visualized vessels in the abdominal cavity. Subsequently, urgent angiography was performed, which did not reveal any contrast in the celiac trunk. At the same time, a significant increase in the levels of Alanine Aminotransferase (ALT), Aspartate Aminotransferase (AST), C Reactive Protein (CRP), lactate, Lactate Dehydrogenase (LDH), amylase and lipase was observed (Figure 2).

Several further procedures were carried out in the following days—their exact description is presented in the Table 1.

On day 3 from the first surgery, relaparotomy was performed, and the partially necrotic spleen was removed. An enlarged pancreas with multiple foci of Balser necrosis was noted. A hematoma of the descending part of the duodenum and an edema of the lower wall of the gallbladder were visible. The drain was placed into the peritoneal cavity near the pancreas. In the following days, the levels of the biochemical markers started to normalize (Figure 2). A significant amount of pancreatic juice leakage through the drain from the abdominal cavity persisted, and conservative treatment with somatostatin at doses of 1.2 mg/day (administered through an infusion pump 24 h a day) did not reduce the drainage volume (1400–1800 mL with a high level of amylase and lipase). From day 17, ischemia and necrosis of the gallbladder, common bile duct, pancreas, duodenum, stomach, and the rest of the spleen were observed. The last surgery was performed on day 59 from the first operation.

Currently, at 141 days after the first surgery, the boy is in stable condition. The last performed MR, CT and MIBG scintigraphy tests, as well as the evaluation of urinary catecholamine excretion, neuron-specific enolase (NSE), and ferritin levels indicate no active neoplastic process. A slight gastric fistula was closed. The child is fed through a jejunostomy tube. The drainage of the common bile duct (with the drain of bile to the outside) is maintained.

Another procedure—anastomosis of the bile duct with the intestinal loop is planned. Due to the over four-month-long interruption in oncological treatment, the decision to continue chemotherapy and/or radiotherapy will be made only if the cancer comes back.

## 3. Discussion

The different ways to treat neuroblastoma include surgery, chemotherapy, radiotherapy, differentiation therapy, immunotherapy, and in selected cases careful observation only. The method of treatment depends on many factors, and both chemotherapeutic and surgical treatment play essential roles in patient therapy [1,2,3].

Since the introduction of Image-Defined Risk Factors (IDRFs) to the International Neuroblastoma Risk Group (INRG) staging system in 2009, their role in predicting surgical complications has been analyzed by many authors [8]. According to the INRG classification, the presented patient was in group L2 [9] (Figure 1).

In order to facilitate the tumor resection, two cycles of chemotherapy were initially applied, resulting in a tumor volume reduction of over 35%. According to the publication by Barak et al., the role of surgery in the treatment of neuroblastoma is evolving [10]. Most patients with the L1 stage can be treated using surgery only. Unfortunately, a gross number of patients are diagnosed at more disseminated stages. In the abovementioned paper, 80% of the patients, including high-risk neuroblastoma tumors, had a gross-total resection of the tumor with minimal operative morbidity and no mortality. In addition, 88% of children had a resection greater than 90% of their lesion, and 3-year survival was 84% [10]. These results confirmed the sense of surgical treatment in our patient. Unfortunately, the postoperative complication related to the ligation of the vessel that was identified as the vessel supplying blood to the tumor resulted in many serious complications. In the angiography examination in the postoperative evaluation, this vessel was surprisingly identified as a celiac trunk.

Celiac artery in adults is a short, broad branch, about 1.25 cm long. The mean arterial diameter of the standard celiac trunk is 0.8 cm [11]. The celiac axis and its branches supply blood to the spleen, pancreas, liver, stomach, and part of the duodenum. The precise knowledge of the localization of the celiac trunk in context of the tumor before the surgery is fundamental [12,13]. It allows for avoiding its damage, vascular surgical complications and patient morbidity.

In the preoperative CT, the celiac trunk diameter in our patient was only 4 mm. The patient presented type 1 of the celiac trunk variation. A mistaken ligation of the celiac trunk may have resulted from the fact that the trunk was compressed and displaced by the tumor.

There is no literature describing the consequences of the ligation of the celiac trunk during surgery in children with cancers. In the publication by Kronick et al., of 2017, information on 11 patients with damage to the visceral trunk due to a blunt abdominal trauma was collected. Most visceral injuries are reported to be the result of penetrating injuries. Overall survival in the described patients was 80%, with one intra-operative death and one death due to fulminant hepatic failure. The author suggests that open surgical ligation is recommended in a haemodynamically unstable trauma patient during damage control operation. The thriving open antegrade surgical bypass was described for hepatic ischemia with ligation alone [14]. The study did not show any other celiac trunk ligation consequences besides a partial disturbance of the spleen blood supply. Margharaby et al. suggested that, in a haemodynamically stable, asymptomatic patient, the treatment options (surgical vs. endovascular repair vs. conservative therapy), reimaging, and observation duration are not as well defined, and they depend on the clinical status of the patient [15]. In the publication by Lim et al., 14 patients with damage to the visceral trunk were mentioned [16]. According to the authors, simple ligation of the celiac trunk is considered superior to surgical reconstruction in achieving hemostasis.

In the literature, possible ligation or occlusion consequences are ischemia of the liver, gallbladder, and spleen [17,18]. It is possible that collateral visceral circulation compensates for the occlusion or ligation of the celiac and superior mesenteric artery. However, mortality rate of these patients was high due to an acute hepatic failure or bowel ischemia. In our patient, we have observed all of the described complications associated with the ligation of the visceral trunk. The boy presented with ischemia and necrosis of the common bile duct, gall bladder, pancreas, and spleen. Transient liver damage was also observed, most likely caused by acute organ ischemia. However, a toxic effect of anesthetic drugs cannot be excluded. Acute pancreatic necrosis and the presence of pancreatic juice in the abdominal cavity were additional factors that contributed to the occurrence of complications such as perforation of the duodenum, gall bladder, and common bile duct.

In the available literature, no such extensive complication as the one that occurred in our patient was described. The only case found in the literature concerned a 75-year-old man with a celiac trunk injury following a motorcycle accident. A pancreatic leak complicated the prolonged hospital course with a pseudocyst formation necessitating ERCP (Endoscopic Retrograde Cholangio Pancreatography) and endoscopic cyst-gastrotomy, as well as with pneumonia, acute kidney injury, and splenic artery pseudoaneurysm managed by exclusion and embolization [14].

To our knowledge, this patient is the first reported case of such an extensive complication after an accidental ligation of the celiac trunk, and the first reported neuroblastoma case to experience such a complication.

## 4. Conclusions

The decision to undergo surgery must consider the operational risk associated with the extent of the tumor. The IDRFs play a significant role in predicting operational risk in the preoperative period in children with neuroblastoma. Detailed imaging diagnostics (CT, MR, angiography) are necessary before deciding on the surgery. Even so, the presence of vascular IDRFs may result in an unexpected complication in the surgical treatment of neuroblastoma in children.

## Figures and Tables

**Figure 1 ijerph-18-01841-f001:**
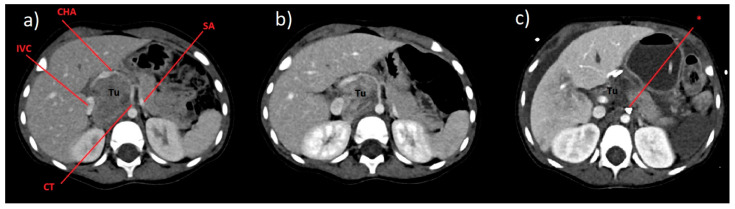
Axial abdominal computed tomography scans in venous phase showed a retroperitoneal tumor in three different examinations of our patient—(**a**) pre-treatment; (**b**) after two courses of chemotherapy; (**c**) postoperative. * in postoperative examination, there was no contrast enhancement of the celiac trunk. In its typical location, the vascular clips were found. Abbreviations: CT—celiac trunk, CHA—common hepatic artery, SA—splenic artery, IVC—inferior vena cava, Tu—tumor.

**Figure 2 ijerph-18-01841-f002:**
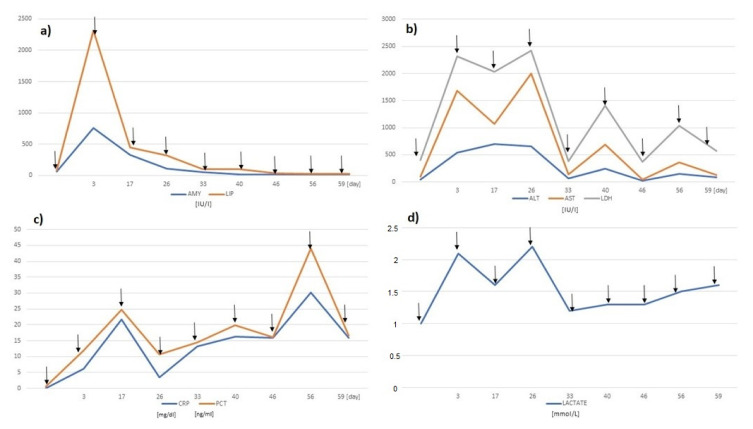
The figure showing the results of biochemical tests in correlation with the performed surgical procedures (marked with arrows). The following tests have been plotted: (**a**) AMY, LIP, (**b**) ALT, AST, LDH, (**c**) CRP, PCT, (**d**) lactate. Normal values for the presented parameters are: AMY 22–80 IU/L, LIP 5–31 IU/L, ALT < 39 IU/L, AST < 48 IU/L, LDH 110–295 IU/L, CRP < 0.5 mg/dL, PCT < 0.5 ng/mL, lactate 0.5–1.6 mmol/L. Abbreviations: AMY—Amylase, LIP—Lipase, ALT—Alanine Aminotransferase, AST—Aspartate Aminotransferase, LDH—Lactate Dehydrogenase, CRP—C Reactive Protein, PCT—Procalcitonin.

**Table 1 ijerph-18-01841-t001:** The table showing the date of the procedure, the imaging test performed, the reason for the intervention, and the surgical treatment used.

Operation Number	Time from First Surgical Intervention (in Days)	Imaging Test Performed	Reason for Intervention	Surgery
1	0	Doppler USG CT	Neuroblastoma	Laparotomy, non-radical resection of the tumor, drainage of the peritoneal cavity, with accidental ligation of the visceral trunk
2	3	Doppler USG Angiography	Spleen ischemia	Laparotomy, partial spleen resection, drainage of the peritoneal cavity
3	17	Doppler USG Angiography MRCP	Ischemia of the gallbladder, pancreas, duodenum	Laparotomy, cholecystectomy, resection of the body and tail of the pancreas, suturing of intraoperative portal vein perforation, drainage of the peritoneal cavity
4	26	Doppler USG CT	Ischemia of the left part of the spleen, hematoma	Laparotomy, resection of the remaining necrotic part of the spleen, removal of the hematoma in the spleen area, drainage of the peritoneal cavity
5	33	Doppler USG CT	Ischemia and perforation of the common bile duct and duodenum	Endoscopy, insertion of a stent into the common bile duct, clip for duodenal perforation
6	40	Doppler USG CT	Re-perforation of the duodenum and common bile duct	Laparotomy, duodenal perforation suturing, pyloric stitching, common bile duct perforation suturing, replacement of the common bile duct stent with a T-drain leading out of the abdomen, anastomosis between the first jejunal loop behind the Treitz ligament and the stomach, drainage of the area where the T-drain is introduced into the bile duct
7	46	CT	Re-perforation of the duodenum and common bile duct	Laparotomy, duodenal perforation suturing, insertion of a Nelaton 4 Fr catheter into the common bile duct with its drainage outside, drainage of the liver hilum area
8	56	CT	Necrosis of the pancreas, duodenum	Laparotomy, pancreatic resection with duodenum (leaving a hook-shaped part), insertion of a Nelaton 8 Fr catheter into the biliary tract, drainage of the liver hilum area
9	59	CT	Stomach ischemia	Laparotomy, gastric sewing, replacement of the Nelaton 8 Fr catheter for the biliary tract under X-ray control

CT—computed tomography, USG—ultrasound examination, MRCP—magnetic resonance cholangiopancreatography.

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
