# Peer review of "The Dramatic Consequences of an Accidental Ligation of the Celiac Trunk during Surgery Performed on a Child with Neuroblastoma"

_ijerph, 2021, doi:10.3390/ijerph18041841_

Round 1

Reviewer 1 Report

This is a report of a patient with intermediate risk neuroblastoma who experienced very serious surgery related complications. The main concern is if this complications could be avoided deciding not to operate the child, considering the operation not essential for the cure of the patient.

The Authors affirm the importance of IDRF in the decision of primary surgery, while it is clear now that IDRF should be considered even for secondary surgery, where the operation is feasible in absence of IDRF, while persistence of IDRF makes difficult the decision of surgery that should depend from the characteristics of every single case. 

Some things to correct:

ABSTRACT

16-19 add "age"; genetic is not only MYCN but also ploidy and 11q study

20 you confound and mix stadiation (L1, L2, M, MS) with risk groups (very low, low, intermediate, high risk)

INTRODUCTION

35-38 and 48-50 details useless for this case. You could delete

50-53 as in the abstract, please add "age" and complete "genetic" with ploidy and 11q

53-54 IDRF is the main aspect of this paper and merits a larger description, particularly its impact on surgery decision

CASE REPORT

64  nothing you indicate in lines 65-70 is suspected for neuroblastoma

70 were

76 please mention here which IDRFs are present in this case (you report it in the Discussion, but here is the right place)

90 trepanobiopsy is not a very usual term; anyway you probably mean bone marrow aspirate and biopsy

92 open or trucut?

98 Please don't mention SIOPEN because SIOPEN protocol is not closed and the patient was not officially recruited

100-101 Surgical decision does not depend on volume reduction but on persistence or not of IDRF

134 No other interventions? It is not true

DISCUSSION

158 Chemotherapy is essential, independently from surgery

163 a gross

165 It is matter of discussion. Persistence of IDRF, particularly in cases of favorable histology, could discourage surgery

187 a haemodinamically

TABLE 1 correct the operation number, there are two "6" 

Author Response

I am very grateful for the review of the article “The dramatic consequences of an accidental ligation of the celiac trunk during surgery performed on a child with neuroblastoma.”  I would like to address your comments and suggestions.

ABSTRACT

  1. 16-19 add "age"; genetic is not only MYCN but also ploidy and 11q study

I am very grateful for that notice.

I added “the age of patient” and delated MYCN. Maybe there is no need to go into detail in the abstract.

I modified it.

The International Neuroblastoma Staging System (INRGSS) is based on the age of patient and preoperative imaging, with attention paid to whether the primary tumor is affected by one or more of specific Image Defined Risk Factors (IDRFs). Patients are classified into the following groups: locoregional L1 and L2 (absent or present IDRFs respectively), M stage (a disseminated form of neuroblastoma) and Ms (the stage present in children younger than 18 months of age with the disease spread to the bone marrow and/or liver, and/or skin). (line 16-22)

I completed information about MYCN in the introduction.

Another factors which influence the prognosis and treatment results are the age of patient, the histopathological type of the tumor, and the presence of genetic markers (MYCN amplification, tumor cell ploidy, the presence of 11q aberration) [6, 7]. (line 60-62)

  1. 20 you confound and mix stadiation (L1, L2, M, MS) with risk groups (very low, low, intermediate, high risk)

Thank you for all your remarks. Please find the corrected version of the Abstract and the Introduction.

ABSTRACT

The International Neuroblastoma Staging System (INRGSS) is based on the age of patient and preoperative imaging, with attention paid to whether the primary tumor is affected by one or more of specific Image Defined Risk Factors (IDRFs). Patients are classified into the following groups: locoregional L1 and L2 (absent or present IDRFs respectively),M stage (a disseminated form of neuroblastoma) and Ms (the stage present in children younger than 18 months of age with the disease spread to the bone marrow and/or liver, and/or skin). (line 16-22)

INTRODUCTION

To minimize complications associated with the surgical procedure, the Image Defined Risk Factors (IDRFs) were introduced in 2009. They have been designed to guide surgical management of neuroblastoma at diagnosis, in particular, to indicate whether a biopsy or attempted resection is recommended as the first surgical procedure. The same system of risk factors can be applied later during treatment and follow-up [6, 7]. Summarizing, it can be said that IDRFs are surgical risk factors and are identified by imaging, based on radiological criteria. For example, abdominal and pelvic IDRFs include: tumor infiltrating porta hepatis or hepatoduodenal ligament, encasing branches of superior mesenteric artery at mesenteric root, encasing origin of celiac axis or superior mesenteric artery, invading one or both renal pedicles, encasing aorta or vena cava, encasing iliac vessels, pelvic tumor crossing sciatic notch [7].

In the International Neuroblastoma Staging System (INGRSS), locoregional tumors are staged L1 or L2 based on the absence or presence of one or more of Image Defined Risk Factors (IDRFs) respectively assessed at the moment of diagnosis. Metastatic tumors are defined as stage M, except for stage MS, in which metastases are confirmed in children younger than 18 months of age to the skin, liver, and/or bone marrow [6, 7].

Another factors which influence the prognosis and treatment results are the age of patient, the histopathological type of the tumor, and the presence of genetic markers (MYCN amplification, tumor cell ploidy, the presence of 11q aberration) [6, 7].  (line 44-62)

INTRODUCTION

  1. 35-38 and 48-50 details useless for this case. You could delete

Thank you very much for your advice.  Sentences “Family history is noted in 1-2% of cases. Neuroblastoma is diagnosed more often in conjunction with other congenital diseases such as Hirschsprung's disease, congenital central hypoventilation syndrom, and neurofibromatosis type 1 [1, 2, 3].” and “The International Neuroblastoma Pathology Classification considers the histopathology results, age of the child, degree of cell differentiation, mitosis-karyorrhexis index (MKI), and presence of Schwann cells [6]” were delated.

  1. 50-53 as in the abstract, please add "age" and complete "genetic" with ploidy and 11q

I am very grateful for that notice.  I completed the necessary information.

I added “the age of patient” and delated MYCN from the abstract.

The International Neuroblastoma Staging System (INRGSS) is based on the age of patient and preoperative imaging, with attention paid to whether the primary tumor is affected by one or more of specific Image Defined Risk Factors (IDRFs). (line 16-22)

I completed MYCN in the introduction.

Another factors which influence the prognosis and treatment results are the age of patient, the histopathological type of the tumor, and the presence of genetic markers (MYCN amplification, tumor cell ploidy, the presence of 11q aberration) [6, 7]. (line 60-62)

  1. 53-54 IDRF is the main aspect of this paper and merits a larger description, particularly its impact on surgery decision

I am very grateful for that notice. A very good point. I increased the description about IDRF.

To minimize complications associated with the surgical procedure, the Image Defined Risk Factors (IDRFs) were introduced in 2009. They have been designed to guide surgical management of neuroblastoma at diagnosis, in particular, to indicate whether a biopsy or attempted resection is recommended as the first surgical procedure. The same system of risk factors can be applied later during treatment and follow-up [6, 7]. Summarizing, it can be said that IDRFs are surgical risk factors and are identified by imaging, based on radiological criteria. For example, abdominal and pelvic IDRFs include: tumor infiltrating porta hepatis or hepatoduodenal ligament, encasing branches of superior mesenteric artery at mesenteric root, encasing origin of celiac axis or superior mesenteric artery, invading one or both renal pedicles, encasing aorta or vena cava, encasing iliac vessels, pelvic tumor crossing sciatic notch [7].

In the International Neuroblastoma Staging System (INGRSS), locoregional tumors are staged L1 or L2 based on the absence or presence of one or more of Image Defined Risk Factors (IDRFs) respectively assessed at the moment of diagnosis. Metastatic tumors are defined as stage M, except for stage MS, in which metastases are confirmed in children younger than 18 months of age to the skin, liver, and/or bone marrow [6, 7].

Another factors which influence the prognosis and treatment results are the age of patient, the histopathological type of the tumor, and the presence of genetic markers (MYCN amplification, tumor cell ploidy, the presence of 11q aberration) [6, 7]. (lines 44-62)

CASE REPORT

  1. 64  nothing you indicate in lines 65-70 is suspected for neuroblastoma\

Thank you very much for your advice. I changed the beginning of the case report part and supplemented the suspicion of the disease in the right place.

A 2.5-year-old boy was admitted to the hospital because he had been lethargic for about two weeks, had experienced chest and abdominal pain, and had been vomiting.  (line 68-69)

On admission to the oncology department, the patient was in a good general condition and showed no abnormality in a physical examination. Imaging examinations – ultrasonography and computed tomography (USG, CT) anteriorly from the right kidney and the thoracolumbar spine showed a solid mass of the tumor, ranging from Th11 to L3 and undergoing significant contrast enhancement. (line 74-78)

  1. 70 were

Thank you very much for that notice. I corrected it.

His parents, older brother, and the closest family were healthy. (line 72-73)

  1. 76 please mention here which IDRFs are present in this case (you report it in the Discussion, but here is the right place)

Thank you very much for your advice. I changed the position of this sentence. 

A neuroblastoma lesion was suspected. Three IDRFs have been identified - the tumor-encased aorta and vena cava, the origin of the celiac axis, and branches of the superior mesenteric artery at mesenteric roots. (line 86-88)

  1. 90 trepanobiopsy is not a very usual term; anyway you probably mean bone marrow aspirate and biopsy

Thank you very much for that notice. I corrected it.

Computed tomography (CT), magnetic resonance (MR), and MIBG scintigraphy, as well as bone marrow aspirate and biopsy excluded the presence of metastases in the lungs, liver, skeletal system, bone marrow, and central nervous system. (line 96-98).

  1. 92 open or trucut?

Thank you for your suggestion. It was an open biopsy. I completed it.

Subsequently, an open biopsy of the tumor was performed. (line 97-98)

  1. 98 Please don't mention SIOPEN because SIOPEN protocol is not closed and the patient was not officially recruited

Thank you very much for that notice. I changed it.

The vascular port was implemented and chemotherapy was applied. (line 102-103)

  1. 100-101 Surgical decision does not depend on volume reduction but on persistence or not of IDRF

Thank you very much for that notice. Yes of course, I changed it.

It was decided to remove the tumor and to continue with further postoperative chemo- and/or radiotherapy. (line 106-107) Why the operation was undertaken was explained in the discussion. (line 166-172)

  1. 134 No other interventions? It is not true

Thank you very much for that notice. Yes of course, it was corrected. I didn't want to repeat all the details of the table.

DISCUSSION

  1. 158 Chemotherapy is essential, independently from surgery

Thank you very much for that notice. Yes of course, it was completed.

The different ways to treat neuroblastoma include surgery, chemotherapy, radiotherapy, differentiation therapy, immunotherapy, and in selected cases careful observation only. The method of treatment depends on many factors, and both chemotherapeutic and surgical treatment play an essential roles in patient therapy [1, 2, 3]. (line 157-160)

  1. 163 a gross

Thank you very much for that notice. I corrected it.

Unfortunately, a gross number of patients are diagnosed at more disseminated stages. (line 168-169).

  1. 165 It is matter of discussion. Persistence of IDRF, particularly in cases of favorable histology, could discourage surgery

Thank you very much for paying attention to this element of the discussion. This is the procedure that we have usually encountered in the literature. And that is also the purpose of determining IDRFs. From our point of view, it was worth showing a different approach to make the discussion of view more attractive. In our case, however, it did not work. After our current experience, with the presence of IDRFs and favorable histology, we believe that it is not worth risking the complications of the surgery. Therefore, the relevant conclusions are included in our publication.

Detailed imaging diagnostics (CT, MR, angiography) are necessary before deciding on the surgery. Even so, the presence of vascular IDRFs may result in an unexpected complication in the surgical treatment of neuroblastoma in children. (line 227-230)

  1. 187 a haemodinamically

Yes, of course it should be “haemodinamically”. Thank you very much for that notice. I corrected it in two places. (line 194 and 198)

  1. TABLE 1 correct the operation number, there are two "6" 

Thank you very much. I corrected it in the manuscript and in the table file.

Once again, thank you for your review and valuable comments.

Reviewer 2 Report

This paper warns of a serious adverse event related to surgery in a patient with intermediate risk neuroblastoma.  The degree of surgical resection needed is debated among risk groups.  I'm not sure that this paper makes sense for this journal however.  I don't know that this is an issue of public health. 

It is an important issue to be evaluated though.  When discussing the staging system, make sure to provide the proper title (INRGSS does not equal international neuroblastoma staging system).

Author Response

Dear Reviewer,

I am very grateful for the review of the article “The dramatic consequences of an accidental ligation of the celiac trunk during surgery performed on a child with neuroblastoma.”  I would like to address your comments and suggestions.

  1. It is an important issue to be evaluated though.  When discussing the staging system, make sure to provide the proper title (INRGSS does not equal international neuroblastoma staging system).

Thank you very much for that notice. It was corrected both in an abstract as well as in the introduction.

ABSTRACT

The International Neuroblastoma Staging System (INRGSS) is based on the age of patient and preoperative imaging, with attention paid to whether the primary tumor is affected by one or more of specific Image Defined Risk Factors (IDRFs). Patients are classified into the following groups: locoregional L1 and L2 (absent or present IDRFs respectively),M stage (a disseminated form of neuroblastoma) and Ms (the stage present in children younger than 18 months of age with the disease spread to the bone marrow and/or liver, and/or skin). (line 16-22)

INTRODUCTION

To minimize complications associated with the surgical procedure, the Image Defined Risk Factors (IDRFs) were introduced in 2009. They have been designed to guide surgical management of neuroblastoma at diagnosis, in particular, to indicate whether a biopsy or attempted resection is recommended as the first surgical procedure. The same system of risk factors can be applied later during treatment and follow-up [6, 7]. Summarizing, it can be said that IDRFs are surgical risk factors and are identified by imaging, based on radiological criteria. For example, abdominal and pelvic IDRFs include: tumor infiltrating porta hepatis or hepatoduodenal ligament, encasing branches of superior mesenteric artery at mesenteric root, encasing origin of celiac axis or superior mesenteric artery, invading one or both renal pedicles, encasing aorta or vena cava, encasing iliac vessels, pelvic tumor crossing sciatic notch [7].

In the International Neuroblastoma Staging System (INGRSS), locoregional tumors are staged L1 or L2 based on the absence or presence of one or more of Image Defined Risk Factors (IDRFs) respectively assessed at the moment of diagnosis. Metastatic tumors are defined as stage M, except for stage MS, in which metastases are confirmed in children younger than 18 months of age to the skin, liver, and/or bone marrow [6, 7].

Another factors which influence the prognosis and treatment results are the age of patient, the histopathological type of the tumor, and the presence of genetic markers (MYCN amplification, tumor cell ploidy, the presence of 11q aberration) [6, 7].  (line 44-62)

Once again, thank you for your review and valuable comments.
